# Aboveground Biomass Productivity Relates to Stand Age in Early-Stage European Beech Plantations, Western Carpathians

**DOI:** 10.3390/plants14192992

**Published:** 2025-09-27

**Authors:** Bohdan Konôpka, Jozef Pajtík, Peter Marčiš, Vladimír Šebeň

**Affiliations:** 1National Forest Centre, Forest Research Institute Zvolen, T.G. Masaryka 2175/22, SK-960 01 Zvolen, Slovakia; bohdan.konopka@nlcsk.org (B.K.); jozef.pajtik@nlcsk.org (J.P.); peter.marcis@nlcsk.org (P.M.); 2Faculty of Forestry and Wood Sciences, Czech University of Life Sciences Prague, Kamýcká 129, Prague 6—Suchdol, CZ-165 21 Prague, Czech Republic; 3Faculty of Forestry, Technical University in Zvolen, T.G. Masaryka 24, SK-960 01 Zvolen, Slovakia

**Keywords:** *Fagus sylvatica*, man-made forest, allometric equations, woody parts and leaves, biomass stock

## Abstract

Our study focused on the quantification of aboveground biomass stock and aboveground net primary productivity (ANPP) in young, planted beech (*Fagus sylvatica* L.). We selected 15 young even-aged stands targeting moderately fertile sites. Three rectangular plots were established within each stand, and all trees were annually measured for height and stem basal diameter from 2020 to 2024. For biomass modeling, we conducted destructive sampling of 111 beech trees. Each tree was separated into foliage and woody components, oven-dried, and weighed to determine dry mass. Allometric models were developed using these predictors: tree height, stem basal diameter, and their combination. Biomass accumulation was closely correlated with stand age, allowing us to scale tree-level models to stand-level predictions using age as a common predictor. Biomass stocks of both woody parts and foliage increased with stand age, reaching 48 Mg ha^−1^ and 6 Mg ha^−1^, respectively, at the age of 15 years. A comparative analysis indicated generally higher biomass in naturally regenerated stands, except for foliage at age 16, where planted stands caught up with the naturally regenerated ones. Our findings contribute to a better understanding of forest productivity dynamics and offer practical models for estimating carbon sequestration potential in managed forest ecosystems.

## 1. Introduction

European beech (*Fagus sylvatica* L.) is one of the most widespread and ecologically significant tree species in the temperate forests of Europe. Beech-dominated forests cover an estimated 15 million hectares across the continent, with the highest concentrations in France and Germany, as well as extensive areas in the southeastern mountainous regions, including the Carpathians, Dinaric Alps, and Balkan Mountains [1,2]. Beyond its broad distribution and considerable commercial value [3], European beech plays a vital ecological role, particularly in maintaining forest biodiversity [4]. Moreover, European beech is increasingly viewed as a promising species for temperate European forests, potentially serving as a resilient alternative to more vulnerable conifers such as Norway spruce (*Picea abies* (L.) Karst.) under shifting environmental conditions [5,6,7].

In Slovakia, European beech occupies approximately 30.1% of the total forested area—about 643,000 hectares [8]. This represents the largest share among all tree species in the country. Data from the Green Report [9] further indicate that the proportion of beech in the national forest stock has increased from approximately 30% to nearly 35% over the last two decades. Notably, nearly one tenth of beech-dominated forest areas are categorized as first-age-class stands (i.e., younger than 20 years). Importantly, European beech has so far shown relatively high resistance to abiotic stressors and pest outbreaks [10,11].

In recent years, closer-to-nature forest management practices have gained traction across Europe [12], emphasizing, among other principles, the use of natural regeneration over artificial planting. Although European beech demonstrates strong potential for natural regeneration under Slovak conditions [13], limitations such as site-specific constraints or insufficient seed availability from maternal stands often necessitate supplementary planting or complete artificial regeneration. Consequently, man-made beech stands remain an important component of forest management in Slovakia.

A major difference between natural and artificial regeneration lies in the initial stand density. Planted seedlings are typically established at densities of several thousand individuals per hectare [14], whereas successful natural regeneration following a mast year can produce extremely dense seedling layers, often exceeding several tens of thousands or even a few hundred thousand individuals per hectare [15]. For example, Konôpka et al. [16] documented as much as 500,000 seedlings per hectare in a three-year-old, naturally regenerated beech stands grown in full sunlight without maternal canopy cover.

These stark contrasts in initial stand density lead to notable differences in early growth dynamics—particularly in crown development and competition for light, water, and nutrients [17]. Such structural and ecological differences influence not only fundamental tree traits like height-to-diameter ratio [18], but also key ecological functions, including net primary productivity (NPP). Given that NPP serves as a vital metric for assessing both wood production potential and carbon sequestration in forest biomass [19], its quantification is essential across all developmental stages of forest stands.

Direct destructive measurements of tree biomass are both costly and time-consuming. For this reason, the development of accurate statistical models is essential. Such models provide a practical and efficient basis for estimating biomass without the need for extensive destructive sampling. Although the traditional log-log transformation and linearization in tree biomass model estimation is still widely used [20,21,22], it proposes a data transformation to reduce the inherent heteroscedasticity. However, this technique often introduces bias in back transformation [23]. Therefore, new techniques, such as weighted non-linear models are increasingly used, as they eliminate the need for data transformation and back-correction [24].

Since NPP is assumed to be related to forest age, and this factor has often been ignored in future NPP projections, more scientific attention should be given to this relationship [25]. He et al. [26] identify two primary limitations in forest age-based NPP models: the lack of spatially explicit forest age datasets and the challenges associated with quantifying NPP–age relationships across diverse forest types and tree species.

However, estimating total NPP—particularly its belowground component—remains a significant challenge. Fine roots, which represent a dynamic and ephemeral component of belowground biomass, are difficult to sample and assign to individual trees especially due to their high turnover [27]. Brunner et al. [28], for example, reported that fine roots (<2 mm diameter) in beech forests exhibit a turnover rate comparable to foliage, slightly over one time per year. In contrast to leaf biomass, the inaccessibility and rapid cycling of fine roots complicate accurate measurements. Although fine roots make an important contribution to total NPP, we were not able to measure them reliably in our study due to these technical constraints. Consequently, we focused exclusively on aboveground NPP (hereafter ANPP) as a more feasible proxy for productivity in forest ecosystems (see also [29,30,31]).

Therefore, this study aims to

(i).Develop models for estimating the aboveground biomass of young, planted beech trees using basic tree characteristics;(ii).Model aboveground biomass accumulation in relation to stand age;(iii).Quantify ANPP and leaf litter productivity regarding stand age;(iv).Compare results from artificially regenerated beech plantations with those obtained from naturally regenerated stands shown in Konôpka et al. [16].

We believe that our findings contribute valuable knowledge regarding the carbon sequestration potential of artificially established beech stands under conditions representative of the Western Carpathians. Moreover, when juxtaposed with data from naturally regenerated stands, this study provides insights—albeit limited to early developmental stages—into the ongoing question of which regeneration method more effectively promotes carbon uptake and storage in European beech forests.

## 2. Results

In 2020, our research represented stands ranging in age from 2 years (Myjava and Zdychava) to 12 years (Cierny Balog; Table 1). Since measurements were conducted in five-year intervals, the maximum observed age extended to 16 years. The plots within each stand had, excepting Cierny Balog, the same age. At the same time, they showed only slight differences in mean tree height and mean stem basal diameter d_0_.

Tree-level models based on 111 sample trees indicated that biomass allocation in relation to stem basal diameter d_0_ or tree height revealed that the aboveground woody parts had nearly seven times more biomass than the leaves (Figure 1). For example, the model showed that beech trees with a stem basal diameter d_0_ of 60 mm had around 400 g of leaves and nearly 3000 g of woody biomass. At the same time stem basal diameter d_0_ was a better predictor than tree height for both woody biomass and foliage (see statistical characteristics in Table 2). The mean square error (MSE) of models using both predictors were lower (0.05–0.09) than of the models where each predictor was used individually. Furthermore, the models using Equation (4) (predictor d_0_) showed better performance with MSE~0.12–0.20 than models using height; MSE~0.39–0.71 (Equation (5)). These results are also supported by the Akaike’s Information Criterion (AIC) and Residual Standard Error (RSE) in Table 2. The results of both metrics clearly demonstrate that combining predictors yields superior performance relative to using them separately.

Moreover, we found that stand age was closely related to both mean stand height and mean stem basal diameter d_0_ (Figure 2). These relationships enabled us to proceed with further calculations using a combined approach: tree-level models based on tree height and stem basal diameter, followed by a stand-level model using age as a predictor.

Then, we calculated, at the plot level, both annual diameter and height increments (Figure 3). We used pooled averages for three age classes: up to 5 years, 6–10 years, and 11–15 years. The results showed small increments in plots aged up to 5 years—about 4 mm for diameter and 0.15 m for height. In plots aged 6–10 years, the increments were slightly larger, while those aged 11–15 years showed nearly 2.5 times greater diameter increment (9 mm) and over 3.5 times greater height increment (0.55 m) compared to the youngest age class. Hence, we observe larger differences in height increment than in diameter increment over time. In fact, differences in both increments among age classes were statistically significant (one-way ANOVA; *p* < 0.001).

Both biomass stocks, i.e., foliage (Figure 4a) and aboveground woody parts foliage (Figure 4b), increased with stand age. Woody biomass stock was significantly greater than that of foliage. For instance, in a 15-year-old beech stand, woody biomass was about 48 Mg ha^−1^, while foliage biomass was approximately 6 Mg ha^−1^. These findings were compared with our previous results focused on young beech stands originating from natural regeneration (Figure 4a–c; Table 3). The comparison indicated that biomass stocks were generally higher in naturally regenerated stands than in planted ones. The only exception was in 16-year-old stands, where foliage biomass in planted stands approached, and would likely later exceed, that of naturally regenerated stands, potentially continuing this trend in older ages.

We calculated ANPP in young, planted beech stands for aboveground woody parts (Figure 5a) and aboveground biomass (i.e., foliage and aboveground woody parts together; Figure 5b). Naturally, NPP of foliage would be nearly equal to its biomass stock, reflecting also one-year foliage litter turnover. NPP of woody parts exceeded that of foliage (Figure 4a and Figure 5a), though the difference was smaller than the contrast in their respective stocks. We also compared aboveground woody part NPP (Figure 5a), and total ANPP (Figure 5b) between planted and naturally regenerated stands (Table 3). In very young stands, naturally regenerated trees exhibited much higher NPP across biomass components. However, by age 15–16, NPP values in planted stands approached those in naturally regenerated ones. Thus, at 16 years of age, both stand types exhibited an NPP of approximately 8 Mg ha^−1^ yr^−1^ for foliage and around 20 Mg ha^−1^ yr^−1^ for woody parts. Hence, the total ANPP at 16 years was nearly 28 Mg ha^−1^ yr^−1^ (Figure 5b).

Finally, we expressed the share of foliage relative to aboveground biomass in terms of both biomass stock and ANPP (Figure 6). Since these shares changed only slightly with stand age, we present the pooled compositions across all stands. While the proportion of foliage to aboveground biomass was 14.9%, its proportion relative to ANPP was more than double, at 30.7%. This highlights the significant role of foliage in carbon assimilation into tree tissues, even though its contribution to the static carbon stock (current status) (i.e., current carbon storage) appears relatively minor.

## 3. Discussion

### 3.1. Biomass Stock and Productivity in Planted Stands

Our results confirmed a well-established finding: that aboveground tree biomass can be modeled using basic characteristics such as tree height and/or stem diameter. These predictors have been widely used by many researchers, either individually (e.g., [32,33,34]) or in combination (e.g., [35,36,37]). Stem diameter, usually measured at 130 cm above ground level (d_1.3_), is a fundamental and commonly used predictor in allometric biomass models. In many forest ecosystems, d_1_._3_ alone provides a highly accurate estimate of aboveground biomass [30]. It is also a practical measurement, as it corresponds to a convenient working height for the operator. However, in our case, d_1.3_ could not be used because some trees were shorter than 130 cm. On the other hand, although tree height generally provides less accurate biomass estimates than stem diameter, it is applicable regardless of tree size [36].

Our tree-level biomass models showed that aboveground woody components (stem and branches) dominated over foliage by approximately a factor of seven. In fact, several previous studies on European beech biomass modeling did not include foliage (e.g., [32,38]) or merged it with branches (e.g., [30,39]), which limited the options for comparing our results with those of other researchers. Bartelink [40] developed allometric relationships for European beech, separately for foliage, branches, and stem. However, the author used d_1_._3_ as the predictor variable, making his models incompatible with ours. Nevertheless, his results showed a gradual decrease in the proportion of foliage to aboveground biomass within the d_1_._3_ range of 5 to 35 cm. For instance, Le Goff and Ottorini [34] reported that in beech stands in France, the contribution of foliage to total tree biomass decreased with age classes. They showed that in stands older than 40 years, leaf biomass did not exceed 5% of total tree biomass (equivalent to about 7% of aboveground biomass). Similarly, Barna and Kodrík [41] found in Slovakia that for 100-year-old beeches, foliage contributed nearly 2% to total tree biomass (i.e., up to 2.5% of aboveground biomass).

Our study demonstrates that weighted nonlinear models provide a reliable framework for estimating tree biomass in young, even-aged stands of different regeneration origins. By incorporating stem basal diameter, height, and their combination as predictors, we achieved improved model performance compared to traditional approaches, particularly in addressing heteroscedasticity without the need for data transformation.

Nevertheless, some limitations remain. The model was constructed using a single dataset and has not yet been validated in independent stands or across broader ecological gradients. Therefore, further studies are necessary to verify its applicability. Despite these limitations, our model represents an important step toward developing efficient, non-destructive methods for biomass estimation and provides a basis for future applications in forest management and carbon accounting.

While tree-level biomass modeling is relatively common, far fewer studies have focused on stand-level equations [42]. This is despite the fact that such models are essential for large-scale biomass predictions, which are crucial for estimating forest carbon storage and dynamics. As a result, our models for stand-level aboveground biomass stock and ANPP in planted beech stands could not be directly compared with other published sources. However, we were able to compare our current results with those from equally young, but naturally regenerated beech stands investigated in our previous study [16] (see next paragraph).

### 3.2. Differences Between Planted and Naturally Regenerated Stands

A comparison between planted beech stands (new data) and naturally regenerated stands (previously published in Konôpka et al. [16] revealed that both foliage and aboveground woody parts NPP were higher in naturally regenerated stands than in plantations. These differences were most pronounced in very young stands, gradually diminishing with age. By 16 years of age, the productivity of both stand types was nearly equal. At this age, ANPP in both planted and naturally regenerated stands was approximately 28 t ha^−1^ yr^−1^, corresponding to about 14 t of carbon sequestered in aboveground biomass annually. In our earlier study [43], the ANPP of a 16-year-old naturally regenerated beech stand was 20 t ha^−1^ yr^−1^. This lower value compared to the current study may be attributed to site-specific conditions, particularly the higher elevation of the previous stand, located at nearly 1000 m a.s.l. The findings suggest that planted beech stands may experience a disadvantage in early developmental stages, likely due to wider initial spacing and transplant shock.

In Slovakia, European beech is typically planted at densities ranging from 3500 to 8000 trees per hectare [14,44]. Under particularly unfavorable site conditions, planting densities can reach up to 10,000 trees per hectare (e.g., [45]). Our field measurements indicated that the actual average stand density in the studied plantations was approximately 4000 trees per hectare, corresponding to a spacing of 1.6 × 1.6 m. Based on this, it can be hypothesized that if beech were planted at a denser spacing of 1.1 × 1.1 m (i.e., slightly over 8000 trees per hectare), the ANPP might be nearly double that observed at our sites. In such a scenario, the ANPP of planted stands could potentially match that of naturally regenerated stands by the age of 12–13 years. However, denser plantations would likely experience earlier canopy closure compared to more sparsely planted stands, which could lead to earlier onset of intraspecific competition and associated tree mortality. This would likely result in earlier and greater carbon losses due to biomass turnover. On the other hand, some studies (e.g., [45]) have reported that higher planting densities in beech plantations tend to reduce the proportion of individuals with poor stem form.

The term “transplant shock” encompasses a range of physiological stress responses in planted trees [46]. It is typically manifested as either increased mortality or reduced growth following planting. In our study sites, tree mortality was negligible; however, growth increments—both in height and diameter—were very low in the youngest stands (up to approximately five years old), indicating that transplant shock likely affected early stand development. From the age of about ten years, beeches in artificial regeneration already had a significantly greater diameter and height growth, so they likely might match or even surpass the productivity of beeches from natural regeneration. Some authors (e.g., [47]) have suggested that methods of planting, planter techniques, and site preparation and maintenance can help reduce transplant shock. This is especially important now, given the negative inherent effects of climate change.

ANPP represents the annual amount of carbon stored in tree biomass, thereby contributing directly to carbon sequestration, i.e., the removal of carbon dioxide from the atmosphere and its storage in biological systems [48]. While carbon stored in woody biomass remains sequestered for several decades, foliage—due to its annual turnover—stores carbon for only a few months before being shed. Once shed, leaves contribute to necromass and continue to store carbon for a short period until fully decomposed. Consequently, the ratio of foliage to woody biomass plays an important role in determining carbon cycling dynamics [49].

Our results showed that foliage accounted for approximately 15% of the total aboveground biomass stock, but its share of ANPP was more than twice that proportion. This finding underscores the significant role of foliage in carbon assimilation, even though its contribution to long-term carbon storage is relatively minor. A similar pattern is likely in naturally regenerated young stands, although the foliage-to-biomass ratio may be somewhat lower due to more intense competition (see [34]). On the other hand, greater carbon loss through tree mortality can be expected in naturally regenerated stands [16], while no mortality due to intraspecific competition has yet been observed in the planted stands we studied. This highlights the need for long-term studies to comprehensively quantify carbon sequestration and cycling in young beech stands established through different reforestation methods.

To fully understand differences in biomass productivity, allocation, and associated carbon dynamics, it is essential to include belowground biomass in future analyses. Our earlier work [50] revealed differences in root-to-shoot ratios between naturally regenerated and artificially planted beech trees, suggesting that growth allocation patterns—both above- and belowground—may differ substantially depending on the way of stand establishment. We suppose that the differences in productivity and biomass allocation between man-made and naturally regenerated stands are related to initial stand density, as well as other factors that may be discovered in future research.

## 4. Materials and Methods

### 4.1. Selection of Stands and Tree Measurements

Before conducting our fieldwork, we coordinated with the State Forest authorities to identify suitable sites and forest stands. These authorities assisted us in the preliminary selection process. The criteria for selecting the sites and stands were as follows:(i).To cover a range of locations across Slovakia, with a focus on moderately fertile sites,(ii).The contribution of beech had to be nearly 100%, with minimal admixture of other tree species,(iii).All trees had to be planted, with no individuals originating from natural regeneration,(iv).The stands had to be established with regular spacing between trees and be up to 12 years old (excluding the 2–3 years the seedlings spent in the forest nursery prior to planting),(v).Seedlings had to be either undamaged or only minimally affected by harmful agents (e.g., browsing by game).

During the 2020 growing season, we selected fifteen stands with European beech (Table 4; Figure 7). These sites were predominantly characterized by Dystric Cambisols, with some occurrences of Stagno-gleyic Cambisols and Eutric Cambisols—soil types typical of, and broadly representative for, a large portion of the Western Carpathians. The sites were situated at elevations ranging from 360 m to 1130 m above sea level, with an average of approximately 700 m.

The average spacing of the planted trees was 1.6 × 1.6 m, corresponding to a mean tree density of nearly 4000 trees per hectare. In each stand, we established three rectangular plots, each encompassing four to five rows of planted trees. To ensure independent sampling units, the distance between individual plots was at least 20 m. Plot sizes varied according to stand density but included at least 30 trees each, with a mean plot area of approximately 85 m^2^.

All trees within the plots were labeled with metal tags bearing unique codes. We recorded their heights and stem diameters d_0_. We measured d_0_ instead of the conventional diameter because some trees were shorter than 1.3 m. Tree heights were measured with a wooden measuring rod (±1 cm), and stem diameters d_0_ were measured using a digital micrometer (±0.1 mm), recorded twice at two perpendicular angles. Measurements were conducted annually in late autumn from 2020 to 2024, resulting in five measurement campaigns.

### 4.2. Tree Sampling and Quantification of Aboveground Biomass

Besides the annual tree measurements, we conducted tree sampling during the end of the growing seasons (at the time when the growth of the trees was completed) in 2021 and 2022. For this purpose, we selected seven beech stands representing a subset of those used for the annual measurements. First, the stands were ranked by size (based on mean stand height), and every second stand was selected in order of size. Then, about 16 randomly selected trees were sampled in each of the seven stands. The trees were chosen outside the rectangular plots to avoid altering the natural conditions within them. This resulted in a total of 111 beech trees being sampled. The sampled beeches were felled on the ground level. The stem diameter d_0_ and tree height were measured for each tree (Figure 8). Aboveground parts of trees were packed into labeled paper bags and transported to the laboratory at the National Forest Centre in Zvolen for further processing.

Under laboratory conditions, beech leaves were manually separated from the branches, resulting in two distinct components: woody parts (branches and stem) and leaves. After separation, each tree component (i.e., woody parts and leaves) was oven-dried at 105 °C for 4–5 days and then weighed with a precision of ±0.1 g. This procedure provided biomass (dry matter weight) data for both leaves and woody parts of every sampled tree.

To create a mathematical model for biomass calculation, 111 samples of beech were implemented. The model development focuses on calculating the dry mass of individual tree covering two basic components: woody parts and leaves. Due to the small size of the trees, it was not possible to use the diameter *d*_1.3_ as an independent variable. Instead, the diameter *d*_0_ was used. Although models where height is the only independent variable are rather rarely used in scientific works, we implemented this approach since in the youngest developmental stages, height is easier to measure than diameter *d*_0_. Moreover, height can be used to connect models of mature stands with models of initial growth stages.

We tested three functions, where the independent variables were, in sequence, diameter *d*_0_, height *h*, and their combination. Because the dataset exhibited heteroscedasticity—i.e., the variance of the dependent variable increased with the values of the predictors—a weighting procedure was applied following the method described by Dutca et al. [24]. This approach involved a two-step process to calculate the weighting variable:
Procedure 1

Heteroscedastic residuals were obtained using an unweighted non-linear model. Residuals (Equation (1)) and predictors were ordered, grouped into sets of 25, and log-transformed (ln-ln). For each group, the mean of the predictor and the variance of the residuals were calculated. A weight for each tree (*wᵢ*) was computed using a linear fit (Equation (2)) of the variance function to calculate the weights (Equation (3)).

Procedure 2

This procedure was identical to Procedure 1, except that the first step involved fitting a weighted non-linear model using the weights derived in the linear fitting step (Equation (4)) of Procedure 1. The parameter λ represents the regression coefficient of the fitted variance model.(1)$$r_{d0i}={BC}_{d0i}-\hat{{BC}_{d0𝚤}}$$(2)$$ln(\sigma_{rd0i}^{2})=\ln\alpha_{i}+\lambda_{i}ln(d_{0i})$$

Here, $r_{D0i}$ is the unweighted residual, ${BC}_{d0i}$ is *i*-th observed biomass component dry weight, $\hat{{BC}_{d0𝚤}}$ is the *i*-th predicted biomass component dry weight, and α and λ are the absolute and regression coefficients estimated from Procedure 2.(3)$$w_{j}=\frac{1}{d_{0j}^{\lambda }}$$

For further details, see Dutca et al. [46], weighting procedures 6 and 7.

The final biomass model equations used in his study took the following forms:(4)$$\hat{{BC}_{𝚤}}=b_{0i}.d_{0}^{b_{1i}}+\varepsilon_{1i}$$(5)$$\hat{{BC}_{𝚤}}=b_{0i}.h^{b_{1i}}+\varepsilon_{2i}$$(6)$$\hat{{BC}_{𝚤}}=b_{0i}.d_{0}^{b_{1i}}.h^{b_{2i}}+\varepsilon_{2i}$$
where *BC*, *d*_0_, and *h* are as previously defined, *b*_0_, *b*_1_ and *b*_2_ are model parameters, and ε represents the random residual term.

If the model failed to converge during Procedure 1, values above the 99th percentile of the response variable were identified as outliers and removed, and the procedure was repeated. The final models were tested for heteroscedasticity using the Breusch–Pagan test and Chi-square statistic [51]. Here the Chi-square *p*-value above 0.05 denies the null-hypothesis of heteroscedasticity present and thus correct heteroscedasticity removal by the weighing procedure. The 95th confidence intervals were calculated using the “propagate” R-package [52]. Second-order Taylor expansion [53] and Monte Carlo simulation were used to construct more realistic error estimates and confidence intervals for nonlinear models than what is possible with only a simple linearization (first-order Taylor expansion) approach. Furthermore, the mean standard error (MSE) was calculated to assess the models’ performance. The models’ development and statistical analysis were performed using the R-statistical software version 4.2.2 [54].

### 4.3. Estimates on Aboveground Biomass Stock and Net Primary Productivity

Based on the beech biomass models from artificial regeneration derived at the tree level for its individual components, the stock of foliage, aboveground woody parts, and total aboveground biomass was calculated for each site (15 sites), its subplot (3 subplots per site), and each year (5 years, i.e., 2020–2024). This was calculated as the sum of biomass stocks of trees growing on the given site and subplot in a particular year. To compare biomass stocks on individual subplots, the stock was recalculated per hectare at a spacing of 1.6 m × 1.6 m in the following way:(7)$$w_{s}=w_{avg}\frac{S}{ab}$$
where
$w_{s}$ is biomass stock per area (Mg per ha);*S* is area size (m^2^);*a*, *b* is tree spacing, i.e., 1.6 m × 1.6 m;
and at the same time:(8)$$w_{avg}=\frac{w}{N}$$
*w_avg_* is average stock of a single tree in the subplot (Mg);*w* is total biomass stock of all trees in the subplot (Mg);*N* is number of trees in the subplot.

NPP of foliage in certain year was calculated as the leaf stock in tones (i.e., Mg) per hectare at a spacing of 1.6 m × 1.6 m. NPP of aboveground woody parts was calculated as the increment (difference between two consecutive years) of woody parts stock of living trees in Mg per hectare at a spacing of 1.6 m × 1.6 m.

The calculated stocks and net primary productions served as input data for building models to estimate the stock of leaves, woody parts, total aboveground biomass, as well as NPP depending on stand age. The models were created using a power function:(9)$$y=b_{0}t^{b_{1}}$$
where
*y* is dependent variable, i.e., biomass stock (Mg per ha) or NPP (Mg per ha and year);*t* is stand age (years);*b*_0_*, b*_1_ are equation parameters.

### 4.4. Comparisons Among Planted and Naturally Regenerated Beech Stands

Since we had previously conducted similar research in naturally regenerated beech stands [16], this provided an opportunity to compare aboveground biomass stocks and ANPP between beech stands established through different methods (man-made versus natural regeneration). The data for natural regeneration were collected from five sites located in Central Slovakia, representing similar soil conditions and altitudes to those in the current study (a detailed description of site and stand characteristics can be found in Konôpka et al. [16]).

The only modification from the original work by Konôpka et al. [16] was the use of a different fitting approach—specifically, power function —to maintain consistency with the method applied to stands originating from artificial regeneration. Both data sources allowed us to make comparisons across stand characteristics: (1) stand stock of woody parts of aboveground biomass (e.g., stems and branches) and foliage, and (2) aboveground net primary productivity (ANPP), combining woody parts and foliage. In both datasets, stand age served as a common predictor, with an overlapping age range between 5 and 16 years.

## 5. Conclusions

We observed that biomass productivity in planted beech stands lagged behind that of naturally regenerated stands during early development, likely due to transplant shock and wider initial spacing. However, this productivity gap narrowed over time, and by age 16, both stand types exhibited similar levels of ANPP. Under Slovak conditions, our estimates suggest that increasing planting density could enhance early growth and allow productivity to approach that of natural regeneration in a little over 10 years. Nevertheless, denser planting may lead to earlier competition and increased tree mortality, potentially resulting in higher carbon loss.

Despite limited early growth in planted stands, mortality remained low, whereas naturally regenerated stands may experience higher mortality due to denser initial spacing. This trade-off highlights the importance of long-term monitoring to comprehensively assess carbon cycling and storage. Additionally, the study emphasizes the need to consider belowground biomass, as root-to-shoot ratios may differ significantly between stand types. The allocation of carbon to foliage, aboveground woody components, and roots ultimately determines the carbon sequestration potential of beech forests.

Here we must note that our stand age–based models for biomass stock and ANPP have their limits. They represent young beech stands on average site qualities in the Western Carpathians. Therefore, they are applicable to this region but may not be suitable for poor or very rich soils. They may also not be suitable for other regions.

Future research should integrate both aboveground and belowground data to better understand the implications of reforestation methods on forest carbon dynamics. Research activities should also extend over longer periods, including the phase after canopy closure. Ideally, experiments would compare natural regeneration with artificial plantations established at varying tree spacing. Such comprehensive studies could provide a valuable knowledge base for practitioners, offering evidence-based recommendations on how to establish beech stands that optimize both biomass productivity and carbon sequestration.

## Figures and Tables

**Figure 1 plants-14-02992-f001:**
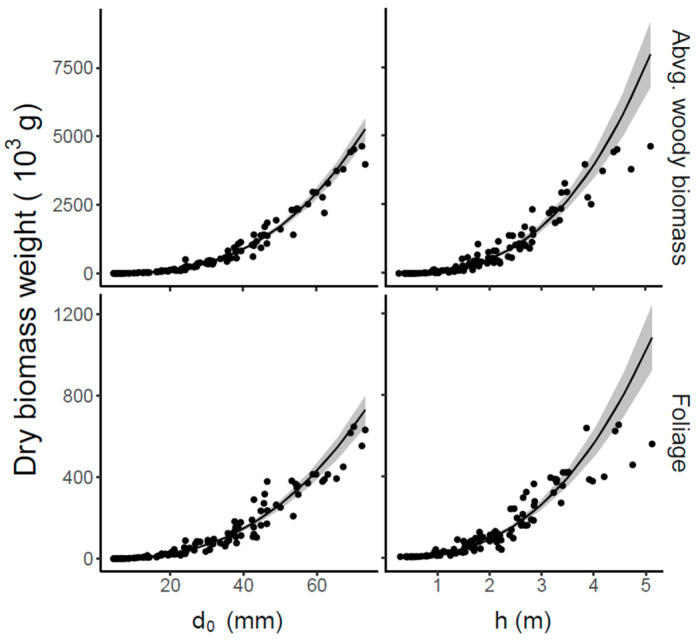
Allometric relationships for aboveground woody biomass (**top panels**) and foliage biomass (**bottom panels**), based on either stem basal diameter (d_0_; **left panels**) or tree height (**right panels**). Corresponding equations and statistical characteristics are provided in Table 2.

**Figure 2 plants-14-02992-f002:**
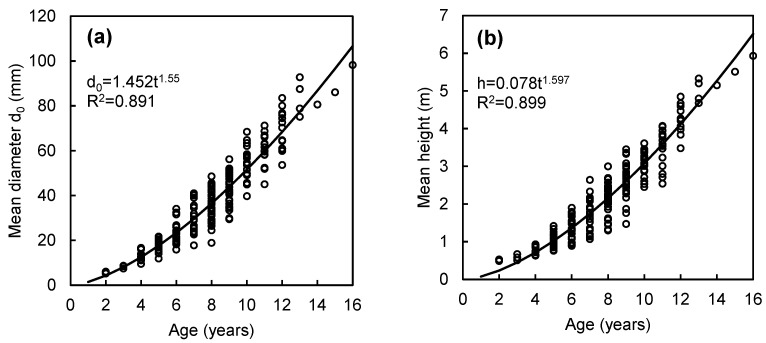
Mean stand basal diameter (**a**) and mean stand height (**b**) in relation to stand age in beech plantations in Slovakia.

**Figure 3 plants-14-02992-f003:**
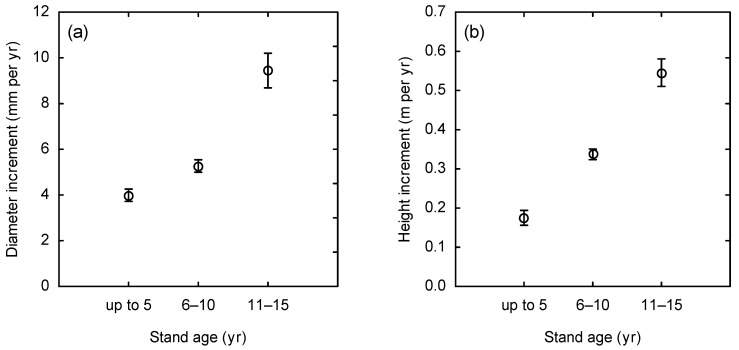
Mean stand diameter (**a**) and mean stand height (**b**) in relation to stand age in beech plantations. The differences between the increments in both cases were significantly important (one-way ANOVA; *p* < 0.001).

**Figure 4 plants-14-02992-f004:**
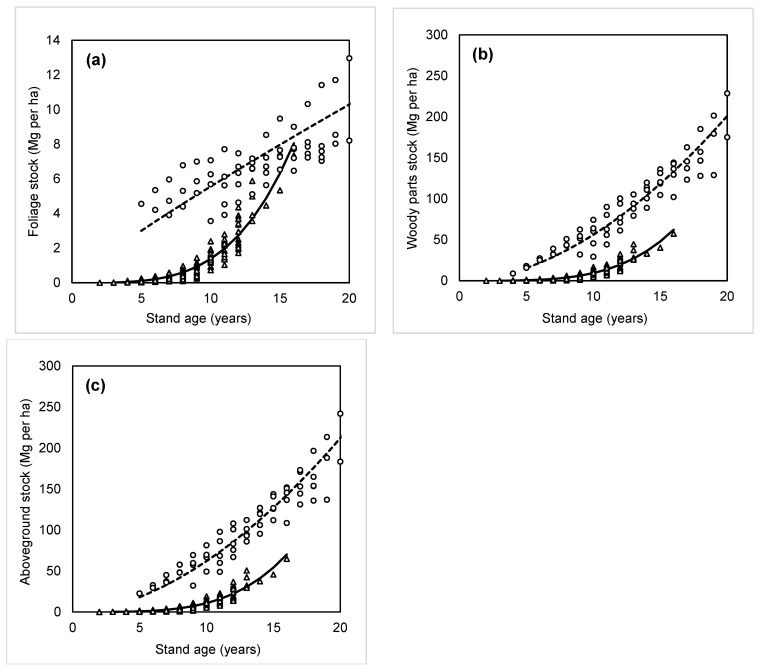
Foliage biomass stock (**a**), aboveground woody parts stock (**b**), and aboveground biomass stock (**c**) in young beech stands by age. Triangles and full lines are dedicated to planted stands, circles and dashed are for naturally regenerated stands. Foliage biomass stock equals both foliage NPP annual litter. Corresponding equations and statistical characteristics are provided in Table 3.

**Figure 5 plants-14-02992-f005:**
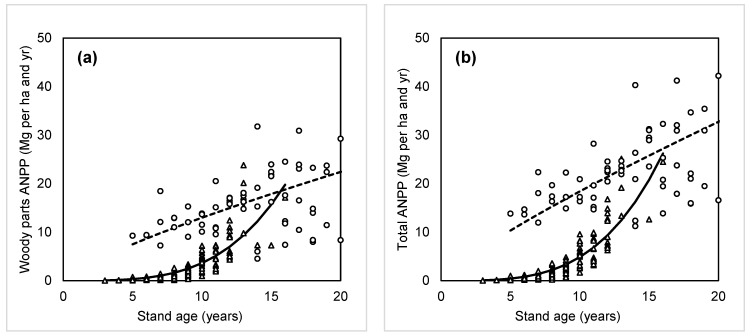
Woody parts ANPP (**a**), and total ANPP (**b**) in young beech stands by age. Triangles and full lines are dedicated to planted stands, circles and dashed are for naturally regenerated stands. Corresponding equations and statistical characteristics are provided in Table 3.

**Figure 6 plants-14-02992-f006:**
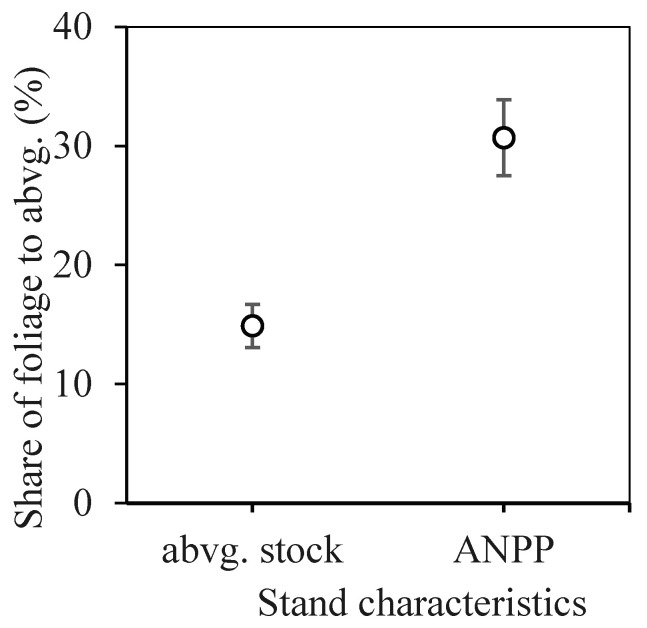
Proportion of foliage in total aboveground biomass, shown for both biomass stock and ANPP, in young planted beech stands. Error bars show standard deviation.

**Figure 7 plants-14-02992-f007:**
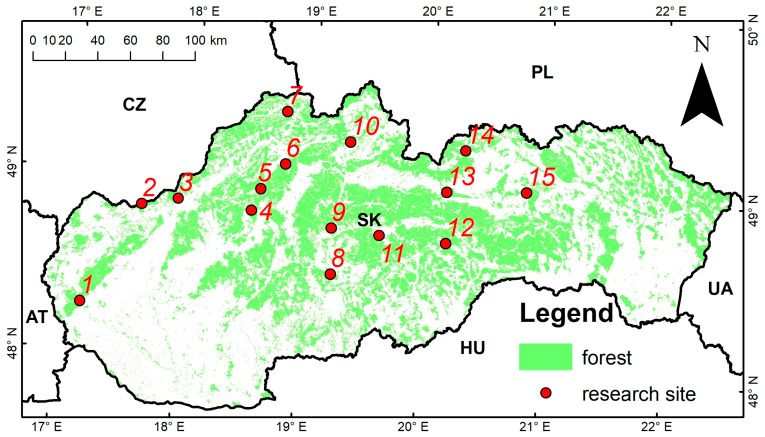
Location of the research sites with planted beech stands selected for measurements, the Western Carpathians, Slovakia (the site codes are explained in Table 4).

**Figure 8 plants-14-02992-f008:**
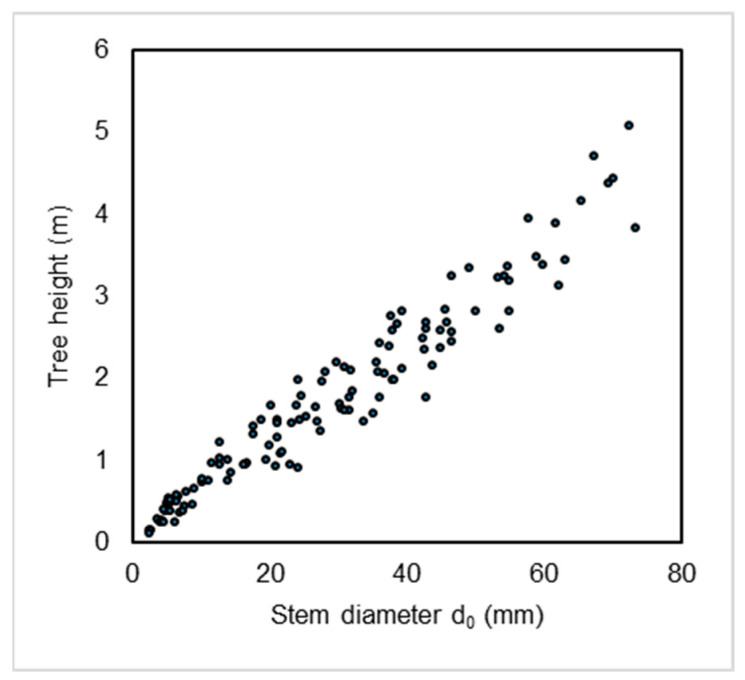
Stem basal diameters (d_0_) and tree heights in beech trees sampled for aboveground biomass modeling (*n* = 111), Western Carpathians, Slovakia.

**Table 1 plants-14-02992-t001:** Stand and tree characteristics of subplots in planted beech stands used for the analysis of biomass stock and ANPP in Western Carpathians, Slovakia (sites are ordered according to stand ages).

Site Name		Number of Measured Trees			Mean Tree Height (m)			Mean Stem Diameter d_0_ (mm)		
	Age (Years)	Subplot A	Subplot B	Subplot C	Subplot A	Subplot B	Subplot C	Subplot A	Subplot B	Subplot C
Zdychava	2	35	34	36	0.26	0.25	0.22	4.52	4.70	4.24
Myjava	2	31	28	33	0.53	0.53	0.49	6.01	5.96	5.17
Svaty Jur	4	34	31	32	0.75	0.63	0.73	12.20	9.47	11.06
Chocholna	4	30	30	30	0.93	0.85	0.91	13.93	12.82	13.90
Poruba	4	30	30	30	0.63	0.63	0.72	16.02	16.12	16.64
Racibor	5	33	33	34	0.89	0.85	1.08	15.47	15.45	18.85
Bijacovce	5	31	31	34	1.23	0.93	1.25	21.49	19.54	19.98
Fackov	6	32	34	32	1.67	1.74	1.90	32.40	31.51	34.07
Zdiar	7	33	31	33	1.77	1.87	1.89	38.97	40.30	40.96
Turie	8	34	32	31	2.12	2.40	2.36	39.75	42.37	41.43
Husarik	8	30	28	30	2.28	2.16	2.34	35.99	36.61	33.56
Lopusna dolina	8	31	31	33	2.27	2.46	2.43	42.84	41.81	38.64
Geberanica	9	29	27	27	1.34	1.18	1.42	18.23	16.78	18.11
Priechod	9	31	24	28	2.68	2.83	2.44	36.34	38.07	34.39
Cierny Balog	10,11,12	30	30	25	2.64	3.84	3.33	35.57	64.66	43.68

**Table 2 plants-14-02992-t002:** Statistical characteristics of tree-level aboveground biomass models based on stem basal diameter (d_0_), tree height (h) and their combination in planted beech stands. Lambda represents the variance function exponent variable. The Chi-squared *p*-value represents the results of Breusch–Pagan heteroscedasticity tests, with *p*-value > 0.05 assuming correct heteroscedasticity removal. df represents the degrees of freedom used in calculation of the Chi-square statistic).

Model	Variable	b_0_ (SE) p_val_	b_1_ (SE) p_val_	b_2_ (SE) p_val_	λ	X2 p_val_	AIC	RSE
f(d_0_)	foliage biomass	0.00878 (0.00138) <0.001	2.63892 (0.04515) <0.001	-	1.821	0.82416	897.2186	0.44905
f(h)	foliage biomass	14.17258 (0.95707) <0.001	2.66097 (0.07205) <0.001	-	1.798	0.73579	997.9132	0.63101
f(d_0_)	abvg. woody biomass	0.01762 (0.0021) <0.001	2.93664 (0.03462) <0.001	-	1.877	0.58194	1211.257	0.34901
f(h)	abvg. woody biomass	68.26919 (5.10207) <0.001	2.92705 (0.07718) <0.001	-	1.764	0.86676	1387.4948	0.84839
f(d_0_,h)	foliage biomass	0.04424 (0.01768) 0.01381	2.03061 (0.14516) <0.001	0.73108 (0.15977) <0.001	1.983	0.55109	876.1298	0.30238
f(d_0_,h)	abvg. woody biomass	0.10546 (0.02987) <0.001	2.27041 (0.10274) <0.001	0.76727 (0.1126) <0.001	1.936	0.52545	1163.9754	0.24043

**Table 3 plants-14-02992-t003:** Statistical characteristics of stand-level models in natural and planted beech stands. The function form is y = b_0_ t^b1^ where b_0_ and b_1_ are parameters and t is stand age. SE means standard deviation, R^2^ is coefficient of determination, and MSE is mean square error.

Origin	Variable	b_0_ (S.E.) *p*	b_1_ (S.E.) *p*	R^2^	MSE
Natural	foliage stock	0.717 (0.167) <0.001	0.889 (0.084) <0.001	0.642	2.458
	woody parts stock	0.802 (0.148) <0.001	1.844 (0.064) <0.001	0.946	250.0
	aboveground stock	1.031 (0.192) <0.001	1.777 (0.064) <0.001	0.940	287.5
	woody parts NPP	2.100 (0.938) 0.028	0.790 (0.161) <0.001	0.287	46.29
	aboveground NPP	2.700 (0.940) 0.005	0.833 (0.125) <0.001	0.423	57.43
Artificial	foliage stock	0.00024 (0.00006) <0.001	3.763 (0.101) <0.001	0.890	0.155
	woody parts stock	0.00084 (0.0002) <0.001	4.044 (0.109) <0.001	0.886	8.93
	aboveground stock	0.00104 (0.0002) <0.001	4.010 (0.108) <0.001	0.886	11.42
	woody parts NPP	0.00086 (0.00048) 0.072	3.622 (0.219) <0.001	0.673	4.53
	aboveground NPP	0.00124 (0.0005) 0.017	3.591 (0.165) <0.001	0.774	4.58

**Table 4 plants-14-02992-t004:** Basic characteristics of research sites with planted beech stands in the Western Carpathians, Slovakia (sites are ordered from west to east; see also Figure 7).

Site Name	Code	Latitude (North)	Longitude (East)	Elevation (m a.s.l.)	Exposition	Slope (%)	Soil Type
Svaty Jur	1	48.2728	17.1855	430	NE	20	Dystric Cambisol
Myjava	2	48.8365	17.6085	550	W	40	Eutric Cambisol
Chocholna	3	48.8833	17.9113	356	NE	40	Rendzina
Poruba	4	48.8527	18.5315	1005	E	30	Dystric Cambisol
Fackov	5	48.9752	18.5959	652	SE	65	Orthic Rendzina
Turie	6	49.1220	18.7869	660	E	65	Eutric Cambisol
Husarik	7	49.4131	18.7688	802	NW	25	Dystric Cambisol
Geberanica	8	48.5329	19.2292	520	NE	45	Eutric Cambisol
Priechod	9	48.7878	19.2094	600	SE	40	Dystric Cambisol
Racibor	10	49.2708	19.3197	674	SW	30	Dystric Cambisol
Cierny Balog	11	48.7648	19.6174	555	-	0	Dystric Cambisol
Zdychava	12	48.7446	20.1758	1130	NW	50	Dystric Cambisol
Lopusna dolina	13	49.0277	20.1591	902	NW	55	Eutric Cambisol
Zdiar	14	49.2621	20.2990	845	NE	60	Dystric Cambisol
Bijacovce	15	49.0498	20.8312	800	NW	20	Dystric Cambisol

## Data Availability

Data are contained within the article.

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
