# Peer review of "Aboveground Biomass Productivity Relates to Stand Age in Early-Stage European Beech Plantations, Western Carpathians"

_plants, 2025, doi:10.3390/plants14192992_

Round 1

Reviewer 1 Report

Comments and Suggestions for Authors

L 24-25: A comparative analysis with naturally regenerated beech stands indicated generally higher biomass in naturally regenerated stands, except for foliage at age.... Please rewrite the sentence.   

General comments:

The authors seek to estimate biomass through the application of allometric models that incorporate height and stem diameter as predictive variables. However, the introduction lacks a thorough discussion on the significance of model-based approaches for biomass estimation. It is essential to elucidate the necessity of employing such models. Utilising data from 15 young, even-aged stands provides a solid foundation; however, models can also facilitate biomass estimations for a wider variety of stand types. The process of manually cutting trees and measuring dry biomass is not only labour-intensive but also costly, underscoring the need for more efficient solutions such as model-based estimations.

The decision to avoid fitting the traditional model to the dataset warrants further examination. A comparative analysis between the traditional model and the modified fitted model is necessary to assess the accuracy of biomass estimation. You utilized various predictors, including tree height, stem diameter, and their combinations. It appears that an ANOVA analysis is absent, which could enhance the comparison of these two models by evaluating degrees of freedom, P-values, AIC, and BIC. Based on the available data, one can conclude that your model demonstrates improved precision in biomass estimation.

Furthermore, it would be beneficial to ascertain whether your updated model indeed provides better fitting and estimates of biomass compared to its predecessor, especially considering the criteria and error metrics employed in fitting the model to the dataset. A reduction of the dataset is advisable to validate the effectiveness of your fitted model within a smaller sample size and within constrained ranges of height, diameter, and dry weight. Alternatively, a calibration approach could be introduced to estimate biomass based on specific diameter measurements.

In the discussion section, it is important to elaborate on both the significance and the limitations of the model for biomass estimation.

In the conclusion section, please write about your model.

The manuscript is well-written, and I do not identify any major issues. It possesses merits that support its publication in this journal.

Author Response

L 24-25: A comparative analysis with naturally regenerated beech stands indicated generally higher biomass in naturally regenerated stands, except for foliage at age.... Please rewrite the sentence.

Thank you, the sentence was rewritten.

General comments:

The authors seek to estimate biomass through the application of allometric models that incorporate height and stem diameter as predictive variables. However, the introduction lacks a thorough discussion on the significance of model-based approaches for biomass estimation. It is essential to elucidate the necessity of employing such models. Utilising data from 15 young, even-aged stands provides a solid foundation; however, models can also facilitate biomass estimations for a wider variety of stand types. The process of manually cutting trees and measuring dry biomass is not only labour-intensive but also costly, underscoring the need for more efficient solutions such as model-based estimations.

The decision to avoid fitting the traditional model to the dataset warrants further examination. A comparative analysis between the traditional model and the modified fitted model is necessary to assess the accuracy of biomass estimation. You utilized various predictors, including tree height, stem diameter, and their combinations. It appears that an ANOVA analysis is absent, which could enhance the comparison of these two models by evaluating degrees of freedom, P-values, AIC, and BIC. Based on the available data, one can conclude that your model demonstrates improved precision in biomass estimation.

Furthermore, it would be beneficial to ascertain whether your updated model indeed provides better fitting and estimates of biomass compared to its predecessor, especially considering the criteria and error metrics employed in fitting the model to the dataset. A reduction of the dataset is advisable to validate the effectiveness of your fitted model within a smaller sample size and within constrained ranges of height, diameter, and dry weight. Alternatively, a calibration approach could be introduced to estimate biomass based on specific diameter measurements.

In the discussion section, it is important to elaborate on both the significance and the limitations of the model for biomass estimation.

In the conclusion section, please write about your model.

The manuscript is well-written, and I do not identify any major issues. It possesses merits that support its publication in this journal.

Thank you for your valuable suggestions. We expanded the Introduction to include a discussion of statistical models and traditional approaches for biomass estimation [L72–79]. Regarding the statistical comparison between the traditional log–log model and our weighted nonlinear model, the ANOVA analysis is not appropriate because the log–log model operates in a transformed scale, whereas the weighted NLS is fitted on the original scale. A comparison using back-transformed predictions for the log–log model and unweighted residuals for the NLS would be possible, but it could also introduce unnecessary bias.

Instead, to evaluate the performance of the weighted NLS models using different predictors (stem base diameter, height, and their combination), we compared Residual Standard Errors and Akaike Information Criterion (AIC) values. This allowed us to identify the best-fitting predictor set. We have updated the Table 2 and added L[130-133]

We acknowledge your suggestion regarding model validation. However, our aim was to construct the most robust model possible using all available data. Since this is one of the first models developed for different regeneration types, we expanded the Discussion and Conclusion to emphasize the need for further validation across additional stands and ecological gradients L[232-242].

Reviewer 2 Report

Comments and Suggestions for Authors

In their manuscript titled “Aboveground Biomass Productivity Relates to Stand Age in Early-Stage European Beech Plantations, Western Carpathians,” Konôpka et al. focused on quantifying aboveground biomass stock and aboveground net primary productivity in young, artificially planted Fagus sylvatica. They further explored differences in carbon storage potential between natural and planted forests. This study is fascinating, well-designed, and reported. I have a few comments as follows.

Uniformly use “stem basal diameter” throughout the text, which helps readers to know the basal diameter was measured instead of 1.3 m height diameter.

Abstract

In the abstract section, authors may condense the methods and expand the results.

Results

Line 112, When the abbreviation MSE appears for the first time, please provide its full English name.

Line 110-119, this section can first describe the data trends in Figure 1, followed by an explanation of the model results in Table 2.

Line 136-137, suggest moving and adding the equation for mean stand basal diameter and height to Figure 2.

Line 154-160, moving this part to the discussion section.

Line 180, these sentences “The function form is: y = b0 tb1 where b0 and b1 are parameters and t is stand age. SE means standard deviation, R2 is coefficient of determination, and MSE is the mean square error.” could be moved to the notes below Table 2.

Discussion

Line 201, when the abbreviation d1.3 appears for the first time, please provide its full English name.

Materials and Methods

Line 296-, for the Materials and Methods section, please add a figure/map to illustrate the 15 sites and their subplots, and show the natural forest stand.

Line 331, change “stem diameters d0” to “stem basal diameters d0

Line 424, change “a b” to “a, b”

References

The authors may consider to add this reference to the sentence in the first paragraph “Beech-dominated forests cover 34 an estimated 15 million hectares across the continent, with the highest concentrations in 35 France and Germany, as well as extensive areas in the southeastern mountainous regions, 36 including the Carpathians, Dinaric Alps, and Balkan Mountains [1,2]” to help showing the distribution of beech in Europe.

Martinez del Castillo, E.; Zang, C.S.; Buras, A. et al. Climate-change-driven growth decline of European beech forests. Commun. Biol. 2022, 5, 163. https://doi.org/10.1038/s42003-022-03107-3

And may consider to add these references to further support the sentence “Moreover, European beech is increasingly viewed as a promising species for temperate European forests, potentially serving as a resilient alternative to more vulnerable conifers such as Norway spruce (Picea abies (L.) Karst.) under shifting environmental conditions [5-7]” in the first paragraph.

Huang, W.; Lundqvist, S-O.; Thygesen, L.G. Effects of climate variability on secondary xylem formation and anatomy in Fagus sylvatica trees grown in Denmark. Bot. Lett. 2025, 172, 87–100. https://doi.org/10.1080/23818107.2024.2426124

Huang, W.; Fonti, P.; Larsen, J.B.; Ræbild, A.; Callesen, I.; Pedersen, N.B.; Hansen, J.K. Projecting tree-growth responses into future climate: A study case from a Danish-wide common garden. Agric. For. Meteorol. 2017, 247, 240–251. https://doi.org/10.1016/j.agrformet.2017.07.016

Author Response

Comments and Suggestions for Authors

In their manuscript titled “Aboveground Biomass Productivity Relates to Stand Age in Early-Stage European Beech Plantations, Western Carpathians,” Konôpka et al. focused on quantifying aboveground biomass stock and aboveground net primary productivity in young, artificially planted Fagus sylvatica. They further explored differences in carbon storage potential between natural and planted forests. This study is fascinating, well-designed, and reported. I have a few comments as follows.

Dear reviewer, thank you for the positive comments.

Uniformly use “stem basal diameter” throughout the text, which helps readers to know the basal diameter was measured instead of 1.3 m height diameter.

Changed in the entire text.

Abstract

In the abstract section, authors may condense the methods and expand the results.

We really wish to do that. However, the maximum length of the text in Abstract is 200. And we have 199. Really sorry. 

Results

Line 112, When the abbreviation MSE appears for the first time, please provide its full English name.

The full name was added.

Line 110-119, this section can first describe the data trends in Figure 1, followed by an explanation of the model results in Table 2.

Done.

Line 136-137, suggest moving and adding the equation for mean stand basal diameter and height to Figure 2.

Done.

Line 154-160, moving this part to the discussion section.

Modified.

Line 180, these sentences “The function form is: y = b0 tb1 where b0 and b1 are parameters and t is stand age. SE means standard deviation, R2 is coefficient of determination, and MSE is the mean square error.” could be moved to the notes below Table 2.

Not quite, because some statistical characteristics are different between the Tables.

Discussion

Line 201, when the abbreviation d1.3 appears for the first time, please provide its full English name.

Done.

Materials and Methods

Line 296-, for the Materials and Methods section, please add a figure/map to illustrate the 15 sites and their subplots, and show the natural forest stand.

The Figure was added. However, we show the 15 plots covering planted stands. Naturally regenerated stands are illustrated in figure within our previous work (Konôpka et al. 2024 – Fig. 1; no repetition is perhaps needed).

Line 331, change “stem diameters d0” to “stem basal diameters d0

Done.

Line 424, change “a b” to “a, b”

Done.

References

The authors may consider to add this reference to the sentence in the first paragraph “Beech-dominated forests cover 34 an estimated 15 million hectares across the continent, with the highest concentrations in 35 France and Germany, as well as extensive areas in the southeastern mountainous regions, 36 including the Carpathians, Dinaric Alps, and Balkan Mountains [1,2]” to help showing the distribution of beech in Europe.

Martinez del Castillo, E.; Zang, C.S.; Buras, A. et al. Climate-change-driven growth decline of European beech forests. Commun. Biol. 2022, 5, 163. https://doi.org/10.1038/s42003-022-03107-3

Yes, we used the suggested citation.

And may consider to add these references to further support the sentence “Moreover, European beech is increasingly viewed as a promising species for temperate European forests, potentially serving as a resilient alternative to more vulnerable conifers such as Norway spruce (Picea abies (L.) Karst.) under shifting environmental conditions [5-7]” in the first paragraph.

Huang, W.; Lundqvist, S-O.; Thygesen, L.G. Effects of climate variability on secondary xylem formation and anatomy in Fagus sylvatica trees grown in Denmark. Bot. Lett. 2025, 172, 87–100. https://doi.org/10.1080/23818107.2024.2426124

Huang, W.; Fonti, P.; Larsen, J.B.; Ræbild, A.; Callesen, I.; Pedersen, N.B.; Hansen, J.K. Projecting tree-growth responses into future climate: A study case from a Danish-wide common garden. Agric. For. Meteorol. 2017, 247, 240–251. https://doi.org/10.1016/j.agrformet.2017.07.016

We used the first suggested citation (from 2025).

Reviewer 3 Report

Comments and Suggestions for Authors

In this study, the authors investigated aboveground biomass and NPP in young European beech plantations.

Estimating forest NPP is thought to be important for estimating the overall forest carbon fixation capacity.

However, I would like to make the following comments:

Introduction

Lines 74-76: Over the years, studies have used litter traps and other methods to estimate leaf biomass production and examine trends in aboveground biomass in stands of various ages, particularly for deciduous trees. Is "Lack of consideration" sufficient for a literature search?

Lines 77-85 mention promising topics such as root biomass below ground, but the next objective states that these will not be considered, which is a discrepancy.

Methods:

Lines 339-349: When was the sampling performed? Leaf biomass, in particular, is affected by the seasonal stage. Wood density also depends on the ratio of earlywood to latewood, so variations in sampling time can be a factor in variations, especially for younger trees.

Line 345 states that n=111 were sampled, but Line 360 ​​states n=120. Please double-check and clearly state what information is correct.

Lines 387-396: The meaning of some abbreviations or parameters, such as 𝑟𝐷0𝑖 and 𝐵𝐶, is not clearly stated. Please clarify.

Lines 431-434: Since NPP is defined per hectare, the information "1.6m x 1.6m" is unnecessary or contradictory.

Results:

Figure 2 would be better divided into separate horizontal figures for diameter and tree height. The symbols overlap and are difficult to distinguish.

Regarding the entire manuscript related to Figure 5, this study focuses solely on aboveground biomass and NPP. Therefore, the term ANPP should be used consistently. If you want to refer to the ANPP of woody parts such as trunks and branches, use ANPP_woody. If you want to refer to leaves, use ANPP_leaf. This will make it easier to understand.

Simply describing woody-part NPP could broadly include woody roots below ground, making the definition unclear given the methodology of this study, which focuses only on aboveground parts.

Discussion:

Lines 236-241: This is related to the purpose of the introduction and the information in Table 1 of the methodology. While this study primarily focuses on stand age after planting, Table 1 also shows significant variation with elevation. As the authors themselves discuss in line L236-, climatic conditions, which are strongly determined by elevation, have a significant impact on forest productivity. Therefore, when interpreting graphs with tree age on the horizontal axis, such as those shown in Figures 2-5, consideration and care must be given to environmental influences such as elevation. Young planted trees are expected to exhibit a "smooth" growth curve, as represented by diameter, provided they are not severely affected by planting shock. However, caution should be exercised regarding the generality and universality of the information in these beautiful graphs.

Author Response

First of all, we would like to thank all the reviewers for their constructive comments. They have helped us greatly in improving the manuscript.

REVIEWER no. 3

In this study, the authors investigated aboveground biomass and NPP in young European beech plantations.

Estimating forest NPP is thought to be important for estimating the overall forest carbon fixation capacity.

Thank you for the positive comments.

However, I would like to make the following comments:

 Introduction

Lines 74-76: Over the years, studies have used litter traps and other methods to estimate leaf biomass production and examine trends in aboveground biomass in stands of various ages, particularly for deciduous trees. Is "Lack of consideration" sufficient for a literature search?

We modified the text.

Lines 77-85 mention promising topics such as root biomass below ground, but the next objective states that these will not be considered, which is a discrepancy.

We changed the text to mitigate the discrepancy.

Methods:

Lines 339-349: When was the sampling performed? Leaf biomass, in particular, is affected by the seasonal stage. Wood density also depends on the ratio of earlywood to latewood, so variations in sampling time can be a factor in variations, especially for younger trees.

We changed the text to explain the situation:

during the end of the growing seasons (at the time when the growth of the trees was completed)…

 Line 345 states that n=111 were sampled, but Line 360 ​​states n=120. Please double-check and clearly state what information is correct.

They were 111 sampling trees (the number is corrected).

 Lines 387-396: The meaning of some abbreviations or parameters, such as ??0? and ??, is not clearly stated. Please clarify.

The full names were added.

 Lines 431-434: Since NPP is defined per hectare, the information "1.6m x 1.6m" is unnecessary or contradictory.

Yes, the NPP is per ha, but we must consider the real spacing as well (since tree density is also important for biomass stock and NPP).

 Results:

Figure 2 would be better divided into separate horizontal figures for diameter and tree height. The symbols overlap and are difficult to distinguish.

Done.

 Regarding the entire manuscript related to Figure 5, this study focuses solely on aboveground biomass and NPP. Therefore, the term ANPP should be used consistently. If you want to refer to the ANPP of woody parts such as trunks and branches, use ANPP_woody. If you want to refer to leaves, use ANPP_leaf. This will make it easier to understand.

Simply describing woody-part NPP could broadly include woody roots below ground, making the definition unclear given the methodology of this study, which focuses only on aboveground parts.

OK, we changed those.

 Discussion:

Lines 236-241: This is related to the purpose of the introduction and the information in Table 1 of the methodology. While this study primarily focuses on stand age after planting, Table 1 also shows significant variation with elevation. As the authors themselves discuss in line L236-, climatic conditions, which are strongly determined by elevation, have a significant impact on forest productivity. Therefore, when interpreting graphs with tree age on the horizontal axis, such as those shown in Figures 2-5, consideration and care must be given to environmental influences such as elevation. Young planted trees are expected to exhibit a "smooth" growth curve, as represented by diameter, provided they are not severely affected by planting shock. However, caution should be exercised regarding the generality and universality of the information in these beautiful graphs.

Yes, altitude is one of the influencing factors, but we are not able to use it in our analyses since only a few sites were included. If we wanted to consider three factors—mode of regeneration, stand age, and altitude—we would need many more sites, which is not feasible under our current conditions. I’m really sorry about that (maybe in our future research). In fact, perhaps soil quality (site class) would even be a stronger factor than altitude itself.

Round 2

Reviewer 3 Report

Comments and Suggestions for Authors

I have confirmed that the authors have updated the manuscript.